# Study on the Optimization and Oxygen-Enrichment Effect of Ventilation Scheme in a Blind Heading of Plateau Mine

**DOI:** 10.3390/ijerph19148717

**Published:** 2022-07-18

**Authors:** Zijun Li, Rongrong Li, Yu Xu, Yuanyuan Xu

**Affiliations:** School of Resources & Safety Engineering, Central South University, Changsha 410083, China; anquan@csu.edu.cn (Z.L.); xy1235813@csu.edu.cn (Y.X.); xuyuanyuan@csu.edu.cn (Y.X.)

**Keywords:** plateau mine, ventilation, oxygen supply, numerical simulation

## Abstract

There are abundant mineral resources in plateau areas, but it is difficult to extract them safely because the problem of hypoxia in plateau mines seriously affects the life and health of workers. In order to address the problem of hypoxia in the blind heading of a plateau metal mine, a three-dimensional roadway model was established based on field data of the Pulang copper mine in Yunnan province, China. The computational fluid dynamics (CFD) method was used to explore the optimal type of oxygen supply duct outlet, and to reveal the oxygen diffusion law influencing different ventilation factors. Grey correlation analysis was used to study the correlation values of the ventilation factors on the oxygen-enrichment effect in blind headings, such as forcing duct position, exhausting duct position, and extraction pressure ratio. The results demonstrated that the oxygen-enrichment effect of a slit oxygen outlet was better than that of the traditional oxygen supply method. When the direction of the oxygen outlet hole was 30° and the height above the roadway floor was 1.95 m, the oxygen increase effect was better than other forms of oxygen supply duct outlets. Grey correlation analysis revealed that the major influencing factors of the oxygen-enrichment effect in the roadway of the plateau mine, were, in descending order, as follows: forcing duct position, extraction pressure ratio, and exhausting duct position. This study has a positive guiding significance for improving the respiration environment in blind headings of plateau mines.

## 1. Introduction

Mineral resources are abundant in the plateau area of western China. With the deepening in development of the Belt and Road policy, the development of mineral resources in western China has been carried out in an orderly manner [1,2]. However, the safety and health of workers in the plateau are greatly affected because of the complex and diverse plateau climate environment that includes low air pressure, low oxygen content, large temperature differences, and other harsh working environments, especially the problem of hypoxia [3]. A plateau mine’s hypoxic environment will cause a series of adverse physiological reactions for personnel who are engaged in the complex physical labors carried out during mining operations. The consequent impacts on labor capacity are particularly obvious; labor capacity decreases by 29.2% at 3000 m, and by 39.7% at 4000 m [4]. Therefore, solving the problem of hypoxia is of great significance, in order to ensure the safety and labor efficiency of workers in plateau mines.

At present, the most common factors for solving the problem of hypoxia in plateau areas are pressurized ventilation and oxygen supply. The equipment required for pressurization technology is very complicated, such as hyperbaric oxygen chambers [5], which are difficult to popularize; their scope and application population are very limited, which makes them more difficult to use in a plateau mine’s complicated operating environment. However, there are abundant basic theories in research about oxygen supply standards [6,7]. Various oxygen supply technologies and methods have emerged, which are of reference significance to the design of mine oxygen supply schemes. The principal oxygen supply methods can be divided into two types: individual oxygen supply and diffusion oxygen supply. The individual oxygen supply method mainly adopts nasal inhalation or mask inhalation, such as using oxygen-increased respirators [8,9]. Oxygen needs to be supplied directly to the nose in order to stimulate the nasal cavity; thus, most personnel need to carry individual oxygen bottles, resulting in additional weight for workers to carry, and increasing their burden [10]. Replacing oxygen tanks can also affect work continuity. This approach is more commonly used in medical institutions. Diffusion oxygen supply is also widely studied. West [11,12,13] analyzed such issues in oxygen-enriched rooms in terms of chamber structure, oxygen concentrators, ventilation levels, costs, fire hazards, and safe upper limits for oxygen enrichment. Barash et al. [14] verified that nocturnal room oxygen enrichment at 3800 m has been shown to reduce the time spent in periodic breathing, and can improve the subjective quality of sleep. Li et al. [15] designed an oxygen dispersal supply system that was a reliable approach to ensuring proper working conditions in high-altitude areas. Li et al. analyzed the oxygen enrichment effect of a small-hole oxygen supply port [16,17] at the end of the oxygen supply duct, and developed a local oxygen supply nozzle [18] to form local oxygen enrichment at the human nose and mouth. Liu et al. [19] proposed a novel point source local diffuse oxygen supply method in order to improve the oxygen environment for sleeping. Despite the considerable amount of research performed by the aforementioned scientists, their established theoretical models do not take into account the influence of the presence of human bodies on roadway airflow, and there is still a lack of comprehensive research on the degree of influence that ventilation factors have on the stability of the oxygen supply effect.

The use of CFD simulations can effectively overcome the problems of time-consuming and costly experiments, and is an effective method for the design of ventilation systems [20,21]. Many scholars at home and abroad have studied the design of mine ventilation systems via CFD simulations, which have been widely used in the study of mine gas extraction and migration laws [22,23], goaf oxidation zone distribution laws [24,25], ventilation pipe layout [26,27,28], and dust migration laws [29,30]. CFD analysis plays an important role in the study of gas diffusion, and can provide a reference for preliminary research carried out prior to laboratory experiments and field verification.

This study aims to provide an optimal ventilation and oxygen supply technical scheme for the improvement of oxygen mass fraction distribution in the blind headings of plateau metal mines. The corresponding numerical model was established using FLUENT software, and the boundary conditions were determined based on the actual conditions of the roadway environment of the Pulang copper mine, at an altitude of 3400 m in Yunnan, the southwest of China. On that basis, the influences of forcing duct position, exhausting duct position, and extraction pressure ratio on the stability of oxygen supply effect were analyzed based on the optimal type of oxygen supply duct outlet. Finally, grey correlation analysis was used to compare the importance of ventilation factors.

## 2. Numerical Analysis

Effective oxygen transfer to the blind heading face of a plateau mine can solve the problem of workers’ hypoxia. There are three types of basic auxiliary ventilation systems: exhaust ventilation, forced ventilation, and hybrid ventilation. However, because of the special environmental conditions of low pressure and hypoxia in the blind headings of plateau metal mines, the use of single exhaust ventilation will easily result in a negative pressure state in the blind heading, which will further reduce the oxygen partial pressure and lead to conditions that are not conducive to mining work. Meanwhile, single forced ventilation will lead to a wide range of smoke diffusion and spread in the roadway along with the airflow, resulting in a low dust removal effect and a poor working environment. Therefore, a hybrid ventilation mode is used in this study.

The mass conservation equation and N–S (Navier–Stokes) equation were used to describe the flow movement in the roadway, as shown in Equations (1) and (2):(1)$$\frac{\partial \rho }{\partial t}+\nabla \cdot (\rho u)=0$$
(2)$$\rho \left(\frac{\partial u}{\partial t}+u\cdot \nabla u\right)=-\nabla p+\nabla \cdot \left(\mu \left(\nabla u+{\left(\nabla u\right)}^{T}\right)-\frac{2}{3}\mu (\nabla \cdot u)I\right)+F$$
where *ρ* is the density of the fluid, kg/m^3^; *t* is the time, s; **u** is the velocity vector of the fluid, m/s; *p* is the pressure of the fluid, Pa; *μ* is the coefficient of kinetic viscosity, Pa·s; **I** is the unit vector; and **F** is the momentum source term, kg/(m^3^·s).

### 2.1. Physical Model

The roadway model was established in FLUENT software, based on the physical conditions of a small cross-section of roadway at the 3400 m level of the Pulang copper mine in Yunnan, the southwest of China. The specific location of the mine is shown in Figure 1. A number of scholars have investigated the environmental characteristics and production situation of the Pulang copper mine, and have taken it as an example to conduct research on roadway ventilation [17,18,31]. The setting of atmospheric pressure, air density, and temperature can be directly referenced by these studies.

Figure 2 shows the basic parameters of the entire roadway (15 m long, 3.0 m wide, 2.3 m high, and 6.3 m^2^ in cross-sectional area) and the position of the duct. The ratio of the geometric model to the actual model was 1:1. Considering the convenience of hanging and saving space, the ventiduct is usually installed on the sidewall or on the top of the roadway. Fresh airflow enters the roadway from the forcing duct, and polluted airflow is diluted and discharged through the exhausting duct. The forcing duct outlet is 10 m away from the heading face, with a diameter of 0.4 m. The height of the center of the forcing duct from the roadway floor is 1.2 m. The outlet of the exhausting duct is 3 m away from the heading face, with a diameter of 0.4 m. The height of the center of the exhausting duct from the roadway floor is 0.5 m. The main working area of the roadway is the space that is located within 6 m from the blind heading face. Most of the traditional oxygen supply methods directly place the oxygen supply duct 6 m away from the blind heading face [16]. For this study, the outlet of the oxygen supply duct is located 6 m away from the heading face, with a diameter of 0.03 m. The height of the oxygen supply duct center from the roadway floor is 1.6 m.

### 2.2. Grid Generation and Independence Tests

In this study, a tetrahedral grid with high fit, high efficiency, and accuracy was selected. Moreover, the grid of the human body area, ducts area, and boundary layer were refined locally, in order to improve the accuracy of the applied model’s calculations. As a result, the total grid number of the model was up to 4,502,986. The geometric model grid after optimization is shown in Figure 3, and the statistics of the skewness and orthogonal quality are shown in Table 1. The maximal skewness was lower than 0.95 and the minimal orthogonal quality was higher than 0.1, suggesting that a high-quality grid had been generated [32].

The efficiency and accuracy of a numerical simulation process are closely related to grid quality, thus establishing grid independence is a necessary condition for analysis [33]. Three different quality grids of low mass (3,319,197 cells), medium mass (4,502,986 cells), and high quality (5,141,688 cells) were selected for independent analysis. The distributions of wind speed and oxygen mass fraction along the roadway were compared, and the relative change rates of the results from different grids were analyzed, as shown in Figure 4. The simulation results for the low−quality grid and medium−quality grid show a higher relative change rate, while the relative change rate in wind speed between the high−quality grid and medium−quality grid is basically less than 5%; the relative change rate of oxygen mass fraction is less than 0.2%. The results show that an increase in the number of grids cannot improve simulation results, indicating that grid independence was achieved. Considering the calculation costs and accuracy, the medium−quality grid method was selected in this study.

### 2.3. Boundary Conditions and Turbulence Model

Before providing oxygen to the roadway, a large air volume should be used to discharge the dust that is generated by blasting, and then turn down the air volume in order to begin providing oxygen to the blind heading face; following this, workers can go into the roadway. At this time, the purpose of ventilation is to discharge the dust generated by workers’ operations. Therefore, the ventilation volume airflow of the forcing duct is determined by the minimum dust exhaust air volume, as shown in Equation (3):(3)$$Q_{0}=V_{0}S$$
where Q_0_ is the volume airflow, m^3^/s, and is also the volume airflow of the forcing duct; V_0_ is the minimum dust removal velocity, 0.15 m/s, in rock roadway; S is the total cross-sectional area of the roadway (blind heading face), m^2^.

According to Equation (3), the minimum airflow volume of the forcing duct is 0.9473 m^3^/s, and the diameter of the forcing duct is 0.4 m; thus, the airflow velocity of the forcing duct is determined to be greater than 7.54 m/s. In this simulation, the airflow velocity in the forcing duct was set at 8 m/s, and the corresponding airflow volume in the forcing duct was 1.0048 m^3^/s. The extraction pressure ratio was 0.6, thus the airflow velocity in the exhausting duct was 4.8 m/s. The airflow volume in the oxygen supply duct was defined as 0.5 m^3^/min, and the corresponding airflow velocity was 12 m/s. The oxygen concentration at the oxygen supply duct outlet was 100%. The boundary conditions of this study are shown in Table 2. The inlet of the forcing duct, the outlet of the exhausting duct, and the inlet of the oxygen supply duct all use inlet velocity, since the airflow velocity inside the duct is known. The velocity and direction of airflow at the roadway exit are unknown; however, the pressure at the roadway exit is known, thus the roadway exit was set as the pressure outlet. The model adopted the steady-state calculation method. Atmospheric pressure, air density, and air temperature were set to the same parameters as the Pulang copper mine roadway model [18].

The k-epsilon turbulence model is suitable for solving exterior flow problems involving complex geometry [26,34], and a number of scholars have experimentally verified that the realizable k-epsilon model is suitable and accurate for the study of oxygen supply ventilation systems [17]; hence, the realizable k-epsilon model is adopted in this paper. The realizable k-epsilon model considers two equation models that deal with turbulent kinetic energy, *k*, and its rate of dissipation, *ε*, which are coupled with turbulent viscosity. This model is represented as follows [32]:(4)$$\frac{\partial }{\partial t}\left(\rho k\right)+\nabla \cdot \left(\rho uk\right)=\nabla \cdot \left[\left(\mu +\frac{\mu_{t}}{\sigma_{k}}\right)\nabla k\right]+G_{k}-\rho \varepsilon$$
(5)$$\frac{\partial }{\partial t}\left(\rho \varepsilon \right)+\nabla \cdot \left(\rho u\varepsilon \right)=\nabla \cdot \left[\left(\mu +\frac{\mu_{t}}{\sigma_{\varepsilon }}\right)\nabla \varepsilon \right]+C_{1\varepsilon }\frac{\varepsilon G_{k}}{k}+C_{2\varepsilon }\rho \frac{\varepsilon^{2}}{k}$$
where *G_k_* represents the generation of turbulence kinetic energy due to the mean velocity gradients, m^2^/s^2^; *C*_1*ε*_ and *C*_2*ε*_ are model constants; *σ_k_* and *σ_ε_* are the turbulent Prandtl numbers corresponding to the *k* equation and the *ε* equation, respectively; *ρ* is the density of the fluid, kg/m^3^; *µ_t_* is the turbulent viscosity, represented by the following:(6)$$\mu_{t}=\rho C_{\mu }\frac{k^{2}}{\varepsilon }$$

The values of *C*_1*ε*_, *C*_2*ε*_, *C_µ_*, *σ_k_*, and *σ_ε_* are 1.44, 1.92, 0.09, 1, and 1.3, respectively.

### 2.4. Optimization Scheme of Oxygen Supply Duct Outlet

A slit oxygen outlet is used to supply oxygen to the respiratory zone in the working area, as shown in Figure 5a. This paper takes the height of 1.0 m to 1.6 m above the roadway floor as the respiratory zone, which represents the main breathing space for a mine worker’s body. The oxygen supply duct is extended to the position of 2.5 m from the heading face, and an oxygen outlet hole 1 m long and 0.01 m wide is arranged on the side near the miner. As shown in Figure 5b, the direction of the oxygen outlet is adjusted so that there is a high concentration of oxygen spray in the human respiratory zone. The positions of four different oxygen outlet holes were analyzed based on the oxygen enrichment effect, in order to compare with traditional oxygen supply methods and obtain the best layout for the oxygen outlet hole. Specific parameter settings are shown in Table 3, and all other conditions used were the same in each calculation model. Case 1 is the traditional oxygen supply mode, as described in Figure 2.

## 3. Results and Discussion

At an altitude of 3500 m, an oxygen mass fraction of 24.3% can effectively relieve the symptoms of hypoxia [16]. The respiratory oxygen mass fraction should be at least higher than 24.3%. The simulation results for the above four cases are shown in Figure 6, and the relevant explanations are as follows:Case 1: when the oxygen was ejected through the outlet of the oxygen supply duct, the velocity direction of oxygen flow was towards the blind heading face, and was affected by the airflow in the roadway. A high concentration of oxygen accumulated on the left wall of the roadway, failing to diffuse sufficiently to the human body. The oxygen mass fraction around the human body ranges from 23.6–23.8%, and the effect of oxygen enrichment is not ideal.Using a slit air outlet, the distribution range of oxygen in the working area of the roadway increased, and the oxygen mass fraction increased significantly, with more concentrated oxygen around the human body. Case 3 presented the optimal result among the five comparisons, with a larger distribution area of high-concentration oxygen; hence, it was applied in the following study.With an increase in oxygen supply duct height, oxygen mass distribution presented two stages. The concentration area of concentrated oxygen moved from the left side of the roadway to the right side, and the distribution range of high-concentration oxygen first increased and then decreased. All these phenomena are related to the airflow driven by forcing ducts. When the oxygen supply duct is lower in height, it is closer to the forcing duct, which is easily affected by the entrainment effect by the jet of the forcing duct, diffusing air to the tunneling face. When the height of the oxygen supply duct is too high, it is difficult to spread oxygen to the floor due to the extrusion of the airflow.

## 4. Stability Analysis of Oxygen-Enrichment Effect

The distribution of oxygen in the roadway is closely related to the ventilation environment [18]. Therefore, this study considered the influences of forcing duct position, exhausting duct position, and extraction pressure ratio on the oxygen-enrichment effect, calculating the oxygen distribution under the action of different influencing factors. The space that is 1 m to 6 m away from the blind heading face is the main working area, thus the oxygen mass fraction in this area along the roadway is analyzed specifically.

### 4.1. Forcing Duct Position

The forcing duct position directly affects the organization form of fresh airflow, and hence the distribution of oxygen. Therefore, the effects of the height between the forcing duct and the roadway floor, in addition to the distance between the forcing duct outlet and blind heading face on oxygen enrichment were considered in this study.

#### 4.1.1. Height between Forcing Duct and Roadway Floor

In order to study the effect of the height between forcing duct and roadway floor on the local oxygen-enrichment effect, heights of 0.7 m, 0.9 m, 1.1 m, and 1.3 m were selected for the calculations, with the other setting conditions kept consistent with Case 3.

As shown in Figure 7, the oxygen mass fractions of the four cases were significantly different, which indicates that the height of the forcing duct above the roadway floor has a significant impact on the oxygen-enrichment effect. With an increase in the height of the forcing duct, the distribution range of high-concentration oxygen (the red part in Figure 7) first increased and then decreased, and gradually diffused to the floor and blind heading face. When fresh air flows out of the forcing duct, the air velocity at the outlet of the jet flow is greater than that of the surrounding air, hence the entrainment effect of the jet flow on the surrounding and air oxygen flow will occur, and the air volume and jet section in the jet fluid will increase continuously. The change in the height between the forcing duct and roadway floor directly affects the entrainment effect and the air distribution in the roadway, and thus affects the oxygen distribution.

A distribution curve for oxygen mass fractions in the main working area along the roadway was made, as shown in Figure 8a. When the height between the forcing duct and roadway floor is 1.1 m, the oxygen mass fraction along the roadway is higher than that at other heights. The variations in oxygen mass fractions along the roadway in the three simulated groups at heights of 0.7–0.9 m are roughly the same, while the variations in oxygen mass fractions along the roadway in the two simulated groups at heights of 1.2 m and 1.3 m are similar. The distributions of the average oxygen mass fractions in the respiratory zone and the entire working area are shown in Figure 8b. The change trends in oxygen mass fraction in the entire workspace and in the respiratory zone are roughly the same, except that when the height is 0.8 m, the oxygen mass fraction in the entire workspace is the lowest, indicating serious oxygen loss.

#### 4.1.2. Distance between Forcing Duct Outlet and Blind Heading Face

As shown in Figure 9, a change in the distance between the forcing duct outlet and the blind heading face made the oxygen distribution in the roadway change, obviously. With an increase in the distance, a high concentration of oxygen tended to move to the back end of the roadway. These phenomena were closely related to ventilation return air. When the airflow out of the forcing duct outlet reached the driving face, the continuity of the pressurized airflow was affected by the wall, and the direction of the airflow consequently changed. The air that returned after impacting the working surface impacts was again induced by the jet, thus forming a vortex region [35]. The oxygen mass fraction at the vortex was significantly higher than elsewhere. The distance between the forcing duct outlet and the blind heading face directly affects the position of the vortex region, and hence affects the distribution of oxygen in the roadway.

As shown in Figure 10a, when the distance between the forcing duct outlet and the blind heading face is 12 m, the average oxygen mass fraction in the main working area along the roadway is lower than that of other groups. Figure 10b shows that when the distance is 12 m, the oxygen mass fractions in the entire workspace and in the respiratory zone are the lowest. When the distance between the forcing duct outlet and blind heading face is greater than 10 m, the change trend in oxygen mass fraction in the respiratory zone is consistent with that in the entire workspace. However, when the distance is 7–9 m, the oxygen mass fraction in the respiratory zone differs greatly from that of the whole working area, indicating that more oxygen cannot be effectively utilized in the respiratory zone of the main working area.

### 4.2. Exhausting Duct Position

The position of the exhausting duct has a direct influence on the airflow field and pressure in the roadway, and thus influences the oxygen distribution in the blind heading face. Therefore, the effects of the height between the exhausting duct and the roadway floor in addition to the distance between the exhausting duct inlet and blind heading face on oxygen enrichment were considered in this study.

#### 4.2.1. Height between Exhausting Duct and Roadway Floor

In order to study the effect of the height between the exhausting duct and the blind heading face on local oxygen enrichment, the heights of 0.3 m, 0.5 m, 0.7 m, and 0.9 m were selected for the calculations; the other setting conditions were maintained consistent with Case 3.

Figure 11 shows that with an increase in the height between the exhausting duct and the roadway floor, the distribution range of high-concentration oxygen first expanded and then decreased. When the height between the exhausting duct and roadway floor reached 0.7 m, the distance between the exhausting duct inlet and the oxygen supply duct outlet was very close, and the oxygen provided by the artificial oxygen supply system could easily be directly pumped out of the driving face, resulting in a poor oxygen-enrichment effect. Figure 12a depicts that the variation trends and law of oxygen mass fraction along the roadway in the main working area can be divided into two categories: the group with a height range of 0.3–0.6 m is similar, and the group with a height range of 0.5–0.7 m is consistent. As shown in Figure 12b, the changing trend in oxygen mass fractions in the entire workspace and in the respiratory zone are roughly the same. When the height between the exhausting duct and roadway floor is 0.6 m, the oxygen mass fraction in the respiratory zone is the lowest, and the difference in the oxygen mass fraction between the entire workspace and the respiratory zone is the greatest, indicating that oxygen cannot be effectively utilized.

#### 4.2.2. Distance between Exhausting Duct Inlet and Blind Heading Face

In order to study the effect of the distance between the exhausting duct inlet and the blind heading face on local oxygen enrichment, distances of 2 m, 4 m, 6 m, and 8 m were selected for the calculations, with all other setting conditions maintained consistent with Case 3.

Figure 13 depicts that with an increase in the distance between the exhausting duct inlet and blind heading face, the distribution area of high-concentration oxygen gradually decreased. This may be because the farther the exhausting duct inlet is from the blind heading face, the stronger the disturbance effect on the airflow in the working area becomes, and the mixed air with a higher oxygen mass fraction is extracted, resulting in a lower oxygen mass fraction in the driving face. As shown in Figure 14a, when the distance between the exhausting duct inlet and the blind heading face is 7 m and 8 m, the average oxygen mass fraction in the main working area along the roadway is lower than that of other groups. Figure 14b shows that as the distance between the exhausting duct inlet and blind heading face increases, the change trend in oxygen mass fraction in the entire workspace and in the respiratory zone is roughly the same.

### 4.3. Extraction Pressure Ratio

The extraction pressure ratio refers to the ratio of the airflow volume in the exhausting duct and the airflow volume in the forcing duct, which reflects a certain effect on the airflow field and pressure in the roadway. Therefore, the influence of the extraction pressure ratio on the ventilation and oxygen supply effect in the heading faces of plateau mines is rigorously studied. Due to the special environment of low pressure and hypoxia in the blind headings of plateau mines, the airflow volume of the exhausting air duct should be less than that of the forcing duct, in order to avoid negative pressure in the roadway which may result in further reductions in partial oxygen pressure. The extraction pressure ratios used were 0.5, 0.6, 0.7, and 0.8, respectively, the airflow volume of the forcing duct remained unchanged, and the corresponding airflow volume and other settings were kept consistent with Case 3.

As shown in Figure 15, with a change in extraction pressure ratio, the oxygen mass fractions of the four groups were similar in the front part of the roadway but significantly different in the back part of the roadway. When the extraction pressure ratio was 0.6, the distribution range of high-concentration oxygen was much smaller than that of other groups. Figure 16a shows that when the extraction pressure ratio is 0.6, the oxygen mass fraction in the main working area along the roadway is the lowest, while the variation trends and law of oxygen mass fractions in the other groups are generally consistent. Figure 16b shows that as the extraction pressure ratio increases, the change trends in oxygen mass fractions in the entire workspace and in the respiratory zone are roughly the same. When the extraction pressure ratio is 0.6, the oxygen mass fractions of the respiratory zone and of the entire workspace are the lowest.

## 5. Degree of Influence on Oxygen Enrichment Factors

In order to facilitate the design of ventilation systems in high-altitude mines, the grey relational analysis method was adopted in this study to further analyze the main factors affecting oxygen distribution in the main working area of the roadway. Grey relational analysis is usually used to determine the influence of various factors on the results; moreover, the quantitative results are consistent with the results of qualitative analyses and have strong applicability. Huang et al. [31] analyzed the specific calculation steps of grey relational analysis in detail.

The oxygen mass fraction in the main working area of the roadway was defined as the system characteristic variable for the grey relational degree model, and the influencing factors on the oxygen mass fraction were defined as related variables. The correlational analysis of variable data was conducted using MATLAB, and the results are shown in Table 4. The main factors that influenced the correlation degree of the oxygen distribution in the blind heading of a plateau metal mine in a descending sequence were as follows: the forcing duct position, the extraction pressure ratio, and the exhausting duct position. The correlation coefficients are all greater than 0.8, indicating that these five factors have a significant impact on oxygen distribution [31]. The positions of the forcing duct and of the oxygen supply duct are on the same side of the roadway, and the airflow provided by the forcing duct has an entrapment effect on the oxygen flow. Therefore, the position of the forcing duct has greater influence on the oxygen distribution in the main working area of the roadway. The results of the grey relational analysis are consistent with the actual situation, which verifies the reliability of the simulation results to a certain extent.

## 6. Conclusions

This paper optimized the outlet form of the oxygen supply duct and also analyzed the effects of different ventilation factors on oxygen enrichment, in order to solve the problem of low oxygen content in the blind heading of a plateau mine and achieve a more comfortable working environment. The results of this study are summarized as follows:Using a slit air outlet, the diffusion and distribution range of oxygen in the working area of the roadway were larger than that of the traditional oxygen supply method, and the oxygen mass fraction increased significantly, with more concentrated oxygen around mine workers’ bodies. When the direction of the oxygen outlet hole is 30° and its height above the roadway floor is 1.95 m, the oxygen-enrichment effect is better than other forms of oxygen supply duct outlets.The forcing duct position, exhausting duct position, and extraction pressure ratio have obvious influence on oxygen-enrichment effects. The influences of these factors on the oxygen distribution are complex and non-monotonous; they initially affect the flow characteristics of air flow in the roadway, hence affecting the oxygen distribution. Based on grey relational analysis results, the main factors that influence the correlation degree of the oxygen distribution in the blind heading of a plateau metal mine, in descending order, are as follows: the forcing duct position, the extraction pressure ratio, and the exhausting duct position. The position of the forcing duct has greater influence on the oxygen distribution in the main working area of the roadway.This paper is mainly a theoretical and feasibility analysis, and can be considered as a preliminary study on oxygen supply in blind headings of plateau mines. In future research, environmental factors and actual working conditions should also be considered, in order to improve the model setting as well as complete the optimization of the artificial oxygen supply system.

## Figures and Tables

**Figure 1 ijerph-19-08717-f001:**
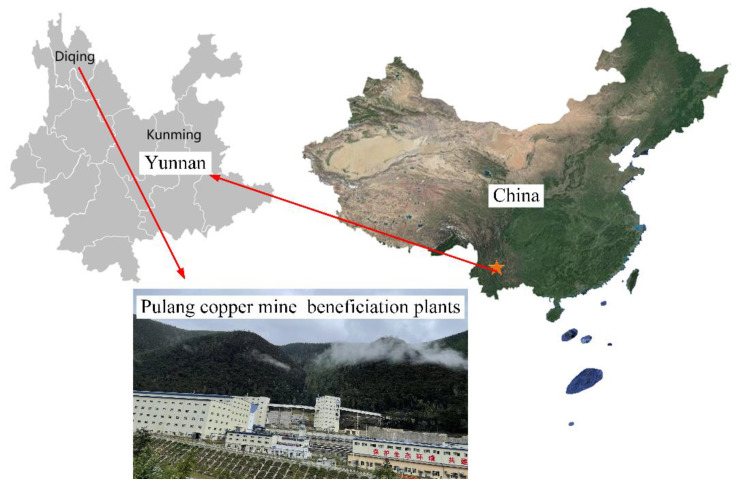
Location of the Pulang copper mine.

**Figure 2 ijerph-19-08717-f002:**
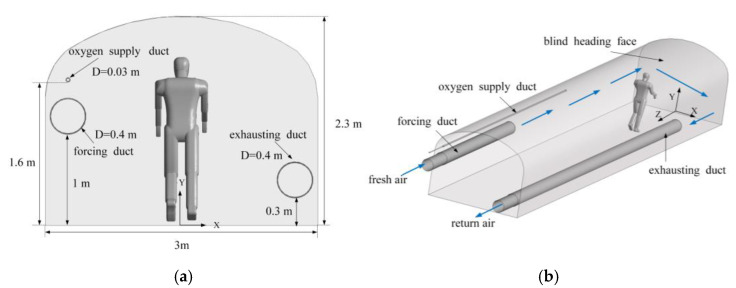
Geometric model of the roadway. (**a**) Cross-sectional size of the roadway; (**b**) three-dimensional model of the roadway.

**Figure 3 ijerph-19-08717-f003:**
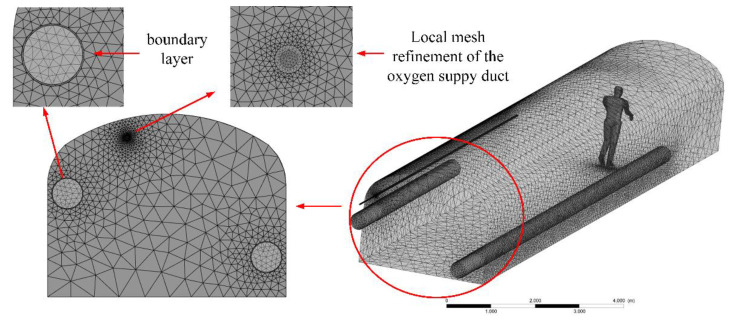
Generation of the roadway grid.

**Figure 4 ijerph-19-08717-f004:**
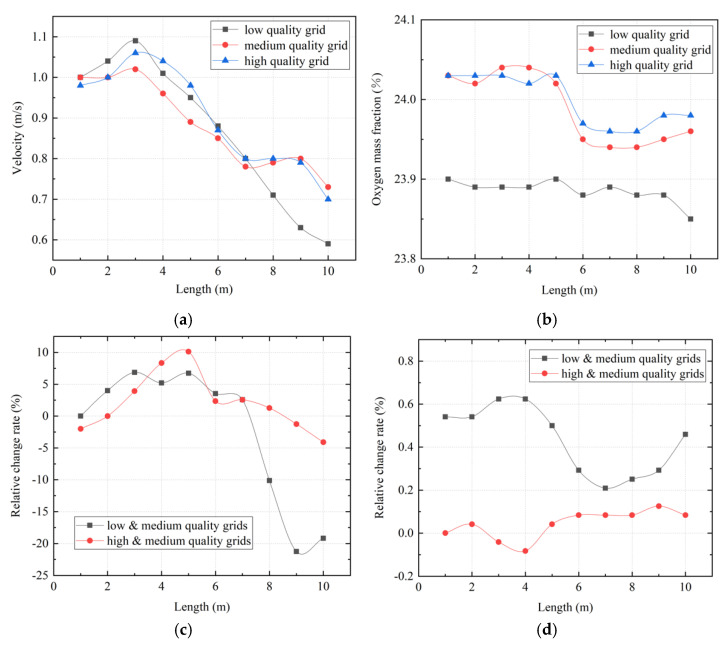
Grid independence analysis. (**a**) Wind speed distribution along the roadway; (**b**) oxygen mass fraction distribution along the roadway; (**c**) relative change rate in wind speed; (**d**) relative change rate in oxygen mass fraction.

**Figure 5 ijerph-19-08717-f005:**
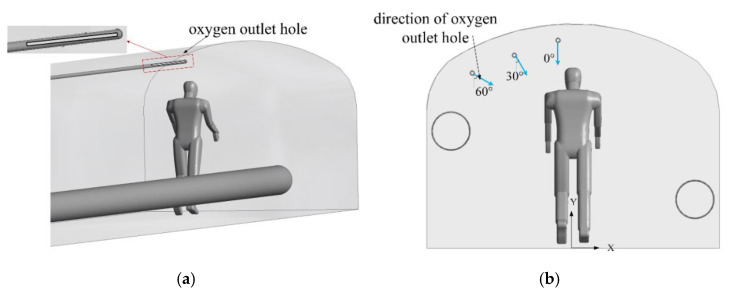
Layout scheme for the oxygen supply duct. (**a**) Outlet of oxygen supply duct; (**b**) directions of oxygen outlet holes.

**Figure 6 ijerph-19-08717-f006:**
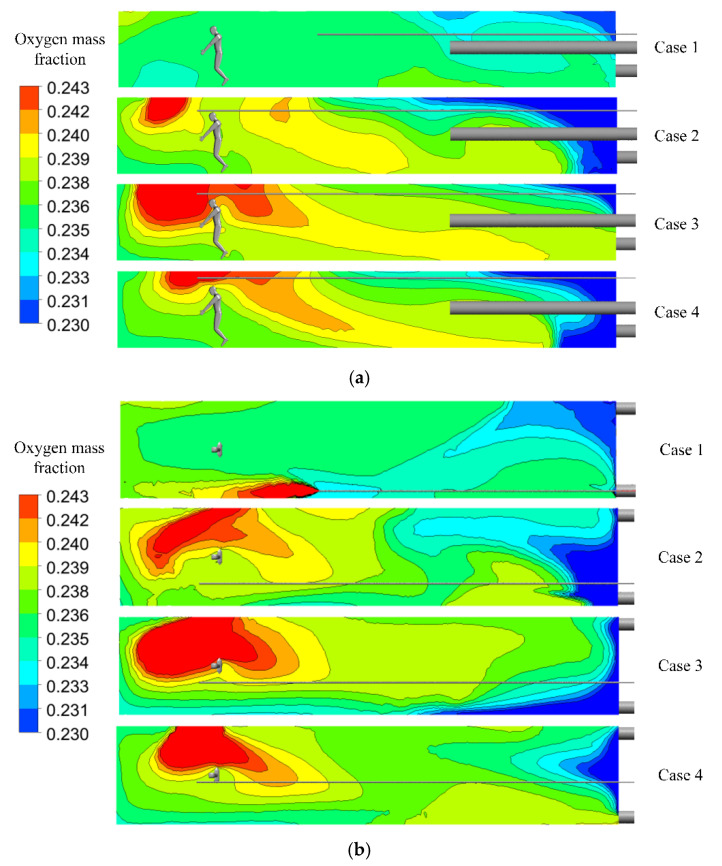
Distributions of oxygen mass fractions. (**a**) Distribution of oxygen mass fractions at section X = 0 m. (**b**) Distribution of oxygen mass fractions at section Y = 1.6 m.

**Figure 7 ijerph-19-08717-f007:**
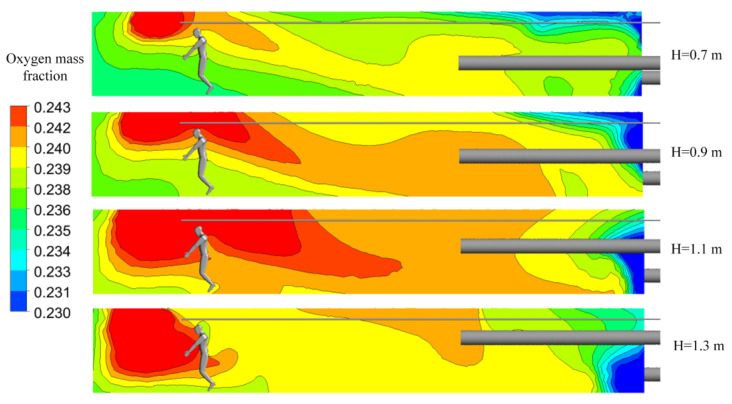
Distributions of oxygen mass fractions at section X = 0 m.

**Figure 8 ijerph-19-08717-f008:**
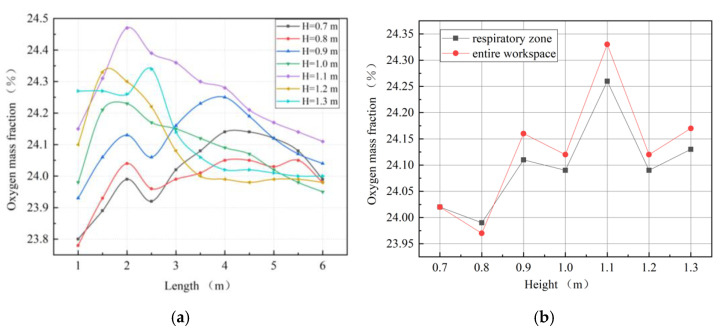
Characteristic diagrams of oxygen mass fraction distributions in main working area. (**a**) Oxygen mass fractions along the roadway; (**b**) oxygen mass fractions in the respiratory zone and in the entire workspace.

**Figure 9 ijerph-19-08717-f009:**
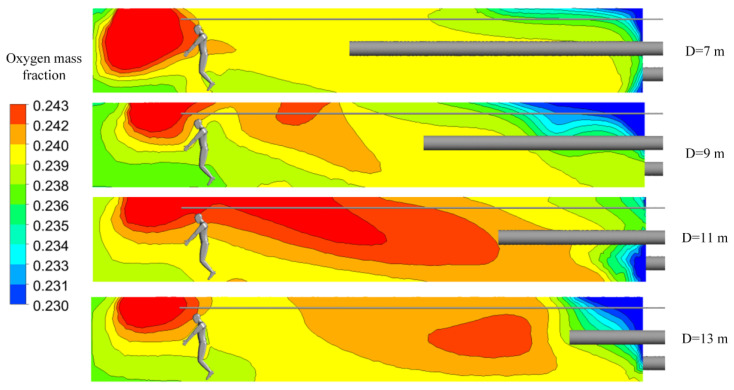
Distributions of oxygen mass fractions at section X = 0 m.

**Figure 10 ijerph-19-08717-f010:**
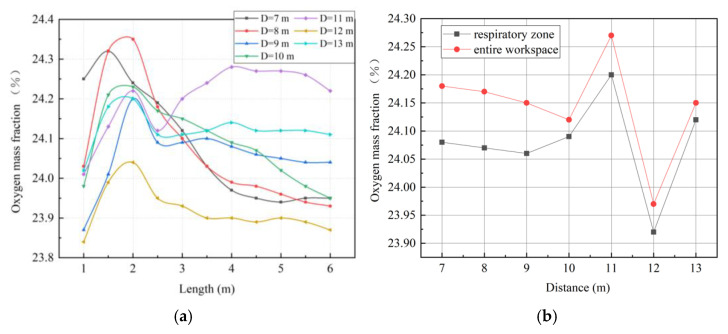
Characteristic diagrams of oxygen mass fraction distributions in the main working area. (**a**) Oxygen mass fractions along the roadway; (**b**) oxygen mass fractions in the respiratory zone and in the entire workspace.

**Figure 11 ijerph-19-08717-f011:**
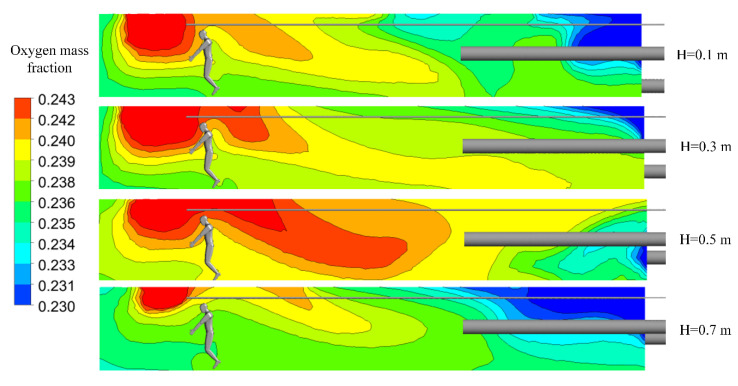
Distributions of oxygen mass fractions at section X = 0 m.

**Figure 12 ijerph-19-08717-f012:**
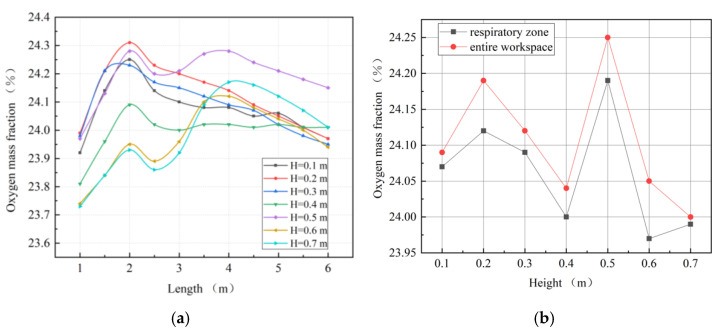
Characteristic diagram of oxygen mass fraction distributions in the main working area. (**a**) Oxygen mass fractions along the roadway; (**b**) oxygen mass fractions in the respiratory zone and in the entire workspace.

**Figure 13 ijerph-19-08717-f013:**
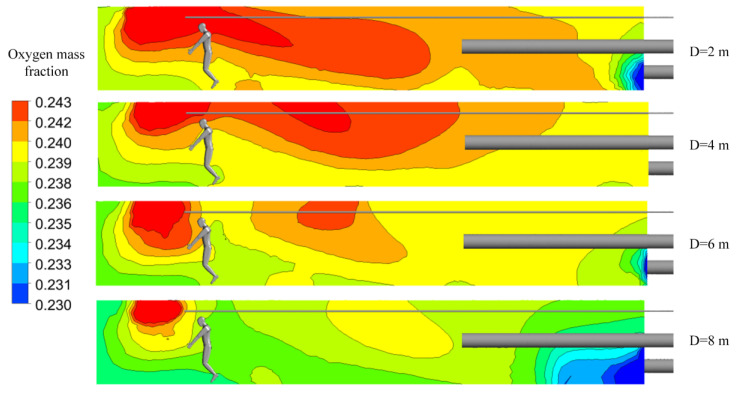
Distributions of oxygen mass fractions at section X = 0 m.

**Figure 14 ijerph-19-08717-f014:**
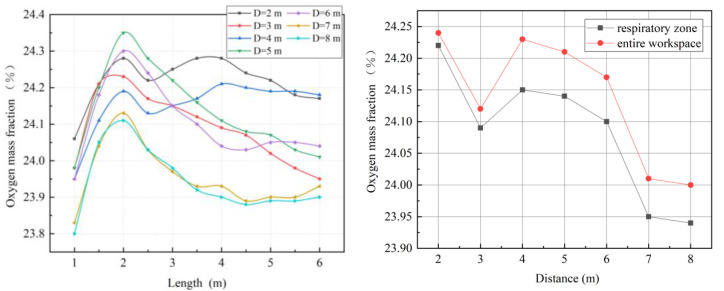
Characteristic diagrams of oxygen mass fraction distributions in the main working area. (**a**) Oxygen mass fractions along the roadway; (**b**) oxygen mass fractions in the respiratory zone and in the entire workspace.

**Figure 15 ijerph-19-08717-f015:**
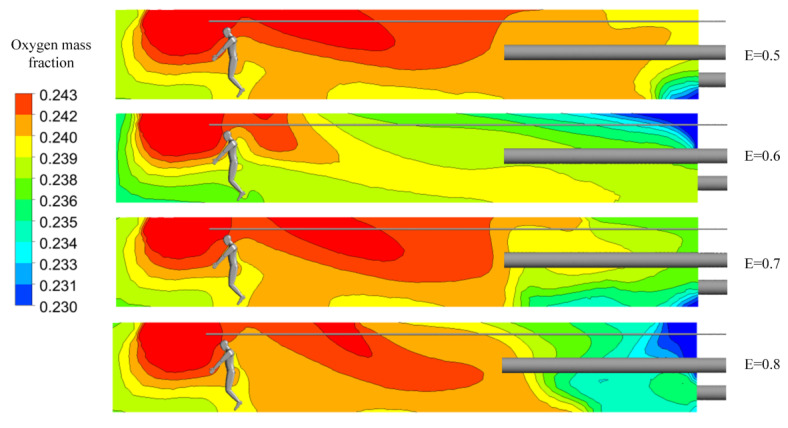
Distributions of oxygen mass fractions at section X = 0 m.

**Figure 16 ijerph-19-08717-f016:**
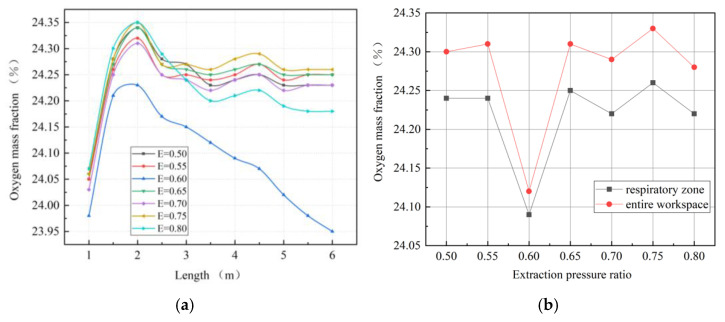
Characteristic diagrams of oxygen mass fraction distributions in the main working area. (**a**) Oxygen mass fractions along the roadway; (**b**) oxygen mass fractions in the respiratory zone and in the entire workspace.

**Table 1 ijerph-19-08717-t001:** Grid metrics information.

Type	Min	Max	Average	Standard Deviation
Skewness	6.89 × 10^−7^	0.90	0.23	0.12
Orthogonal quality	0.18	0.99	0.86	0.08

**Table 2 ijerph-19-08717-t002:** Boundary conditions.

Type	Parameter
Forcing duct outlet	Inlet velocity, 8 m/s
Exhausting duct outlet	Inlet velocity, 4.8 m/s
Oxygen supply duct outlet	Inlet velocity, 12 m/s
Roadway exit	Outflow, 66.614 Kpa
Other walls	Wall
Temperature	288.15 K
Air density	0.8064 kg/m^3^

**Table 3 ijerph-19-08717-t003:** Optimization scheme for the oxygen supply duct outlet.

Case	Height above the Roadway Floor/m	Distance from the Left wall/m	Direction of Oxygen Outlet Hole
Case 2	1.80	0.60	60°
Case 3	1.90	1.00	30°
Case 4	2.00	1.40	0°

**Table 4 ijerph-19-08717-t004:** Calculation results of the correlational analysis.

Influencing Factors	Height between Forcing Duct and Roadway Floor	Distance between Forcing Duct Outlet and Blind Heading Face	Height between Exhausting Duct and Roadway Floor	Distance between Exhausting Duct Inlet and Blind Heading Face	Extraction Pressure Ratio
Correlation coefficient	0.9767	0.9500	0.8191	0.8422	0.8636

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
