# Peer review of "Study on the Optimization and Oxygen-Enrichment Effect of Ventilation Scheme in a Blind Heading of Plateau Mine"

_ijerph, 2022, doi:10.3390/ijerph19148717_

Round 1

Reviewer 1 Report

In order to improve this article, I think it is necessary the following:

-          A list of the annual fatal and non-fatal accidents that have occurred in the blind heading of plateau mines in last 5 years related to hypoxia. This would give a dimension of the current problem that this article aims to reduce.

-       Carry out an analysis on the technical possibility of installing the oxygen supply duct, based on the analyzed extraction system or other possible combinations. As well, how it can be managed with blasting in order to avoid damage.

-        Higher oxygen levels than 21% (such as the situations shown in figures 10, 12 or 14) can create additional risks regarding explosive and inflammable situations. That fact must be deeply commented because it affects the research validity in already hazardous environments.

Author Response

Dear Reviewer:

We sincerely thank you for your letter and for the reviewers’ comments concerning our manuscript entitled “Study on the Optimization and Oxygen-Enriched Effect of Ventilation Scheme in a Blind Heading of Plateau Mine” (Manuscript ID: ijerph-1744262). Those comments are all valuable and very helpful for revising and improving our paper, as well as the important guiding significance to our future studies. We have studied comments carefully and have made corrections which we hope meet with approval. Revised portions are marked in red in the paper. The point-by-point response to the reviewer’s comments are as follows:

(1) A list of the annual fatal and non-fatal accidents that have occurred in the blind heading of plateau mines in last 5 years related to hypoxia. This would give a dimension of the current problem that this article aims to reduce.

Response: We sincerely appreciate the valuable comments. The statistics of related accidents caused by hypoxia in the blind heading of plateau mines are less because the effect of hypoxia on human body is mainly reflected in the weakening of labor capacity. As the altitude rises, oxygen content gradually decreases, leading to a decrease in labor capacity. We supplement the influence of different altitudes on labor capacity in this paper (in 35 and 36 Line).

(2) Carry out an analysis on the technical possibility of installing the oxygen supply duct, based on the analyzed extraction system or other possible combinations. As well, how it can be managed with blasting in order to avoid damage.

Response: Thanks for your suggestion. This paper is basic research on the optimization of the outlet form of oxygen supply duct and the influence of the hybrid ventilation on the stability of oxygen enrichment effect. The selected hybrid ventilation mode is based on the previous research, which is more suitable for heading face of plateau mine [1]. On this basis, the influence of the forcing duct position, exhausting duct position, and the extraction pressure ratio on the oxygen enrichment effect was studied. In addition, after the blasting and dust removal are completed, oxygen supply is then carried out when the miner enters the roadway. At this time, the installation of oxygen supply duct, to avoid blasting damage to the duct.

[1] Li, Z.; Zhao, S.; Li, R.; Huang, Y.; Xu, Y.; Song, P. Increasing Oxygen Mass Fraction in Blind Headings of a Plateau Metal Mine by Oxygen Supply Duct Design: A CFD Modelling Approach. Math. Probl. Eng. 2020, 2020, doi:10.1155/2020/8541909.

(3) Higher oxygen levels than 21% (such as the situations shown in figures 10, 12 or 14) can create additional risks regarding explosive and inflammable situations. That fact must be deeply commented because it affects the research validity in already hazardous environments.

Response: We sincerely appreciate the valuable comments, which have important guiding significance to our future studies. The oxygen content in this study is expressed by the oxygen mass fraction. The oxygen volume fraction in the atmosphere is 20.9%, which is converted to 23.19% by mass fraction. When operating in confined space, the safe range of oxygen content is 19.5% -23.5% (volume fraction), then the oxygen mass fraction is about 21.6% -26.0%. The oxygen mass fraction of the article is in this safe range, follow-up studies should also pay attention to the safe upper limit of oxygen concentration. 

Special thanks to you for your good comments!

Reviewer 2 Report

Observations / point to be considered by Authors:

Good paper with detailed presentation of preliminary modelling studies. 

(1) The exaggerated sudden peaks or dips shown in graphs of figures 8(b), 10(b), 12(b), 14(b) and 16(b) could be misleading. The y-axis scale is exaggerated and the total difference in oxygen mass fraction is only around 0.3% to 0.5% in y axis.

In addition, those peaks or lows shown in the graphs, which are obtained from CFD simulations could be due to the nature of turbulence modelling in CFD simulations. Are you getting exactly the same results (peaks and lows at exactly same distances) if you increase the number of iterations - OR- repeat the simulations ?

(2) CFD modelling studies carried out seems to be too simplistic to develop those firm conclusions  (particularly considering very low difference in oxygen content distribution values). Presentation of these results as 'preliminary results' are OK for paper publication. 

(3) What is the low oxygen content limit allowed in those mines ? you need to specify in the paper. Any legal / mining regulation limits on minimum oxygen content required in respiratory zone ?

(4) What exactly is the objective of this study - with respect to oxygen limit. What is the targeted oxygen content at workers respiratory zone. 

(5) In mining environments, oxygen content can vary around 0.4% at different locations even under normal conditions. 

(6) Why are you using oxygen mass fraction values - rather than the standard mole fraction values ?. Does the regulations in China specify minimum mass fraction values for oxygen content ?

(7) In general oxygen content value is specified as mole fraction value. For example, oxygen content in air is around 21%. The minimum oxygen content limit in some countries is about 19% oxygen (in intake areas where people need to work). 

  You may consider the above points in your revised paper. 

In addition, it would be good for the paper, if you can also present some field measured oxygen content distribution values in the headings. 

Author Response

Dear Reviewers:

We sincerely thank you for your letter and for the reviewers’ comments concerning our manuscript entitled “Study on the Optimization and Oxygen-Enriched Effect of Ventilation Scheme in a Blind Heading of Plateau Mine” (Manuscript ID: ijerph-1744262). Those comments are all valuable and very helpful for revising and improving our paper, as well as the important guiding significance to our future studies. We have studied comments carefully and have made corrections which we hope meet with approval. Revised portions are marked in red in the paper. The point-by-point response to the reviewer’s comments are as follows:

(1) The exaggerated sudden peaks or dips shown in graphs of figures 8(b), 10(b), 12(b), 14(b) and 16(b) could be misleading. The y-axis scale is exaggerated and the total difference in oxygen mass fraction is only around 0.3% to 0.5% in y axis. In addition, those peaks or lows shown in the graphs, which are obtained from CFD simulations could be due to the nature of turbulence modelling in CFD simulations. Are you getting exactly the same results (peaks and lows at exactly same distances) if you increase the number of iterations - OR- repeat the simulations?

Response: We sincerely appreciate the valuable comments. The increase in oxygen levels was limited and difficult, so the increase in oxygen content in the study was only between 1% and 2%. We used the same numerical model of roadway and ventilation system as the previous study, including the setting of some boundary conditions and the selection of turbulence model, and verified the accuracy of the numerical simulation results through experiments in the previous study [1], so this numerical simulation is reliable. In addition, we tried to increase the number of iterations, but the results did not differ.

[1]. Li, Z.J.; Wang, J.J.; Zhao, S.Q.; Xu, Y. The effect of oxygen supply and oxygen distribution on single-head tunnel with different altitudes under mixed ventilation. Indoor And Built Environment 2021, 10.1177/1420326x211051414, doi:10.1177/1420326x211051414.

(2) CFD modelling studies carried out seems to be too simplistic to develop those firm conclusions (particularly considering very low difference in oxygen content distribution values). Presentation of these results as 'preliminary results' are OK for paper publication. 

Response: Thanks for your suggestion. This paper is mainly a theoretical and feasibility analysis of the optimization of the oxygen supply duct outlet form and the influence of the hybrid ventilation on the stability of oxygen enrichment effect, and we will further optimize the oxygen supply duct outlet form by employing laboratory experiments based on the numerical simulation results of this paper.

(3) What is the low oxygen content limit allowed in those mines? you need to specify in the paper. Any legal / mining regulation limits on minimum oxygen content required in respiratory zone?

Response: We sincerely appreciate the valuable comments. At an altitude of 3,500 m, 24.3% oxygen mass fraction can effectively relieve the symptoms of hypoxia [2]. The respiratory oxygen mass fraction should be at least higher than 24.3%. Thank you for your valuable comments, we have supplemented this content (in 235,236 and 237 Line).

[2]GB/T 35414-2017, Requirements of oxygen conditioning for indoor oxygen diffusion in plateau area[S].

(4) What exactly is the objective of this study - with respect to oxygen limit. What is the targeted oxygen content at workers’ respiratory zone? 

Response: Thanks! This study aims to provide an optimal ventilation and oxygen supply technical scheme for the improvement of oxygen mass fraction in the blind heading of Plateau metal mines. The targeted oxygen mass fraction in workers’ respiratory zone should be at least higher than 24.3%.

(5) In mining environments, oxygen content can vary around 0.4% at different locations even under normal conditions. 

Response: We sincerely appreciate the valuable comments, which have important guiding significance to our future studies. It is worth mentioning that, because of the complexity of the actual working environment of the mine roadway, this paper simplified the model to a certain extent, and the simulation was carried out in a more ideal state. This paper can be considered as a preliminary study on oxygen supply in blind heading of plateau mines. In future research work, environmental factors and actual working conditions should also be considered to improve the model setting and complete the optimization of the artificial oxygen supply system.

(6) Why are you using oxygen mass fraction values - rather than the standard mole fraction values? Do the regulations in China specify minimum mass fraction values for oxygen content?

Response: Thanks! The cloud images and data derived from the FLUENT software are oxygen mass fractions, so the oxygen mass fraction is used in this study. The national standard of the People's Republic of China (GB/T 35414-2017) stipulates the requirement of oxygen volume fraction at different altitudes.

The volume fraction and mass fraction of oxygen can be directly converted by the formula (1). The main components of air are oxygen and nitrogen. The mole fraction is equal to the volume fraction. The molar mass of oxygen and nitrogen is 32 g/mol and 28 g/mol, respectively.

                                               32VO/ (32VO+28 VN) = Vm  (1)

where VO represents the oxygen volume fraction, VN represents the nitrogen volume fraction, and Vm represents the oxygen mass fraction. Air can be thought to consist roughly of oxygen and nitrogen.

   The volume fraction of oxygen and the mass fraction of oxygen can be converted from each other through formulas. The oxygen mass fraction can be calculated when the oxygen volume fraction is known.

(7) In general oxygen content value is specified as mole fraction value. For example, oxygen content in air is around 21%. The minimum oxygen content limit in some countries is about 19% oxygen (in intake areas where people need to work). 

Response: Thanks! The oxygen volume fraction in the atmosphere is 20.9%, which is converted to 23.2% of the mass fraction of oxygen. The volume fraction and mass fraction of oxygen can be directly converted by the formula. In previous published articles, oxygen mass fraction has also been used [3-4].

[3]. Li, Z.; Zhao, S.; Li, R.; Huang, Y.; Xu, Y.; Song, P. Increasing Oxygen Mass Fraction in Blind Headings of a Plateau Metal Mine by Oxygen Supply Duct Design: A CFD Modelling Approach. Math. Probl. Eng. 2020, 2020, doi:10.1155/2020/8541909.

[4]. Li, Z.J.; Wang, J.J.; Zhao, S.Q.; Xu, Y. The effect of oxygen supply and oxygen distribution on single-head tunnel with different altitudes under mixed ventilation. Indoor And Built Environment 2021, 10.1177/1420326x211051414, doi:10.1177/1420326x211051414.

(8) In addition, it would be good for the paper, if you can also present some field measured oxygen content distribution values in the headings. 

Response: Thanks for your suggestion, which is helpful for our paper, but we are very sorry for we do not have the field measured oxygen content distribution values. In the future research, we will carry out some field experiments based on the results of numerical simulation to further enhance the oxygen content in blind heading of plateau mine.

Special thanks to you for your good comments!

Reviewer 3 Report

Fig.2 - what is the purpose of such a detailed description of the geometry of a miner? And why is it located in the indicated place of the blind heading? What kind of work does he perform at the same time? Can he stand not in the middle of the cross section?

line 130 - was the prismatic boundary layer built on the walls?

line 148 - what is oxygen mass fraction? How is it calculated and what values ​​are considered acceptable? It's not clear, because usually researchers operate with volume fractions.

Fig. 4 - how can one judge the independence of the solution from the grid from here? What relative residuals were specified for calculations?

lines 146-147 - not grids but cells! a grid is a set of cells, one grid is used for one calculation.

line 156 - Nothing is said about the equations for calculating the velocity and pressure fields, the field of concentration of various gases in the air. Which gas components were taken into account in the calculations? 

In general, a description of the system of equations is usually made first, and then you should say about the numerical grid and the numerical method for solving the equations.

line 169 - it turns out that the air jet at a speed of 8 m/s hits directly into the back of the mine worker. Is this normal in terms of a comfortable working conditions?

equations 2 and 3 - it would be better to depict the speed with a vector arrow.

line 182 - what is the oxygen concentration at the oxygen supply duct outlet? 25%? 50%? 100 %?

line 187 - which k-epsilon model was used? standard, RNG, realizable?

line 236 - where is the cross-section X = 0 located ? Does it correspond to the middle vertical section or not?

It is very difficult to judge the patterns of oxygen transfer from the oxygen supply duct outlet along the selected plane sections. Why didn't the authors consider 3D distributions when visualizing the solution?

Figs 12,14,16. - Dependences of oxygen mass fraction on H, D and E are nonmonotonic. The nature of the obtained dependencies is very complex and not clear from the point of view of physics. The authors do not comment on this, while this is very important. Why should I trust these incomprehensible curves? Maybe the fact is that the solution is unsteady. Then it can contribute and influence the curves in the figures 12, 14, and 16?

Author Response

Dear Reviewers:

We sincerely thank you for your letter and for the reviewers’ comments concerning our manuscript entitled “Study on the Optimization and Oxygen-Enriched Effect of Ventilation Scheme in a Blind Heading of Plateau Mine” (Manuscript ID: ijerph-1744262). Those comments are all valuable and very helpful for revising and improving our paper, as well as the important guiding significance to our future studies. We have studied comments carefully and have made corrections which we hope meet with approval. Revised portions are marked in red in the paper. The point-by-point response to the reviewer’s comments are as follows:

(1) Fig.2 - what is the purpose of such a detailed description of the geometry of a miner? And why is it located in the indicated place of the blind heading? What kind of work does he perform at the same time? Can he stand not in the middle of the cross section?

Response: Thanks! Considering the influence of miners in the roadway on the air flow and oxygen distribution, a human model is added to the physical model, which is also convenient to observe the oxygen concentration in the surrounding respiratory zone of the miners. The movement states of the miners are various when they are drilling and so on. It is difficult to consider all the movement states completely, so it is simplified.

(2) line 130 - was the prismatic boundary layer built on the walls? Fig. 4 - how can one judge the independence of the solution from the grid from here? What relative residuals were specified for calculations?

Response: We sincerely appreciate the valuable comments. The prismatic boundary layer was built on the fluid. To more intuitively analyze grid independence, we supplemented the relative change rates of wind velocity and oxygen mass fraction between the different grids (in 163-167 and 172 Line).

(3) line 148 - what is oxygen mass fraction? How is it calculated and what values ​​are considered acceptable? It's not clear, because usually researchers operate with volume fractions. line 182 - what is the oxygen concentration at the oxygen supply duct outlet? 25%? 50%? 100 %?

Response: Thanks! The cloud images and data derived from the FLUENT software are oxygen mass fractions, so the oxygen mass fraction is used in this study. The oxygen volume fraction of the atmosphere is 20.9%, which is converted to 23.2% of the oxygen mass fraction. In previous published articles, oxygen mass fraction has also been used [1,2]. In addition, the oxygen concentration at the oxygen supply duct outlet is 100% and we have made a supplementary explanation in the article (in 192 and 193 Line).

The main components of air are oxygen and nitrogen. The mole fraction is equal to the volume fraction. The molar mass of oxygen and nitrogen are 32 g/mol and 28 g/mol, respectively. The volume fraction and mass fraction of oxygen can be directly converted by the formula (1).

                                          32VO/ (32VO+28 VN) = Vm  (1) 

where VO represents the oxygen volume fraction, VN represents the nitrogen volume fraction, Vm represents the oxygen mass fraction. Air can be thought to consist roughly of oxygen and nitrogen.

The volume fraction of oxygen and the mass fraction of oxygen can be converted from each other through formulas. The oxygen mass fraction can be calculated when the oxygen volume fraction is known.

[1]. Li, Z.; Zhao, S.; Li, R.; Huang, Y.; Xu, Y.; Song, P. Increasing Oxygen Mass Fraction in Blind Headings of a Plateau Metal Mine by Oxygen Supply Duct Design: A CFD Modelling Approach. Math. Probl. Eng. 2020, 2020, doi:10.1155/2020/8541909.

[2]. Li, Z.J.; Wang, J.J.; Zhao, S.Q.; Xu, Y. The effect of oxygen supply and oxygen distribution on single-head tunnel with different altitudes under mixed ventilation. Indoor And Built Environment 2021, 10.1177/1420326x211051414, doi:10.1177/1420326x211051414.

(4) lines 146-147 - not grids but cells! a grid is a set of cells; one grid is used for one calculation.

Response: We are deeply sorry for our negligence. We have modified this error description (in 159 Line).

(5) line 156 - Nothing is said about the equations for calculating the velocity and pressure fields, the field of concentration of various gases in the air. Which gas components were taken into account in the calculations? In general, a description of the system of equations is usually made first, and then you should say about the numerical grid and the numerical method for solving the equations.

Response: We sincerely appreciate the valuable comments. The mass conservation equation and N-S equation were used to describe the flow movement in the roadway (modify in 102-112 line). The fluid in the roadway is set as air, and the air density is set as 0.8064 kg/m3 according to the altitude. Air is also supplied by the forcing and exhausting ducts. The gas supplied by the oxygen supply pipe is pure oxygen.

(6) line 169 - it turns out that the air jet at a speed of 8 m/s hits directly into the back of the mine worker. Is this normal in terms of a comfortable working conditions?

Response: Thanks! The ventilation volume airflow of the forcing duct is determined by the minimum dust exhaust air volume to discharge the dust generated by miners' operation. The airflow velocity in the forcing duct outlet was set at 8 m/s, but the forcing duct outlet is 10 m away from the heading face. When the airflow provided by the forcing duct flows to the main working area, the average wind speed is generally reduced to between 0.5-2 m/s. This range of wind speeds is comfortable for the human body.

(7) equations 2 and 3 - it would be better to depict the speed with a vector arrow.

Response: Thanks for your suggestion. These two formulas are written and calculated with reference to the relevant literature which adopts the same turbulence model as this study [3,4].

[3] Wang, W.H.; Zhang, C.F.; Yang, W.Y.; Xu, H.; Li, S.S.; Li, C.; Ma, H.; Qi, G.S. In Situ measurements and CFD numerical simula-tions of thermal environment in blind headings of underground mines. Processes 2019, 7, 313, doi:10.3390/pr7050313.

[4] Li, Z.; Zhao, S.; Li, R.; Huang, Y.; Xu, Y.; Song, P. Increasing Oxygen Mass Fraction in Blind Headings of a Plateau Metal Mine by Oxygen Supply Duct Design: A CFD Modelling Approach. Math. Probl. Eng. 2020, 2020, doi:10.1155/2020/8541909.

(8) line 187 - which k-epsilon model was used? standard, RNG, realizable? line 236 - where is the cross-section X = 0 located? Does it correspond to the middle vertical section or not?

Response: Thanks! I am deeply sorry for our negligence. A number of scholars have experimentally verified that Realizable K-Epsilon model is suitable for the study of oxygen supply ventilation system with high accuracy, so the Realizable K-Epsilon model is adopted in this paper. we have made a supplementary explanation in the article (in 188 and 189 Line). In addition, the cross-section X = 0 correspond to the middle vertical section.

(9) It is very difficult to judge the patterns of oxygen transfer from the oxygen supply duct outlet along the selected plane sections. Why didn't the authors consider 3D distributions when visualizing the solution?

Response: Thanks for your suggestion, which is helpful for our paper, but we are very sorry for we do not have 3D distributions. In the process of post-processing the results, we tried to generate 3D distribution map of oxygen mass fraction, but the visualization effect was not good. Different angles were needed to see oxygen distribution at different locations. Therefore, we chose to show oxygen distribution in a cross-sectional diagram.

(10) Figs 12,14,16. - Dependences of oxygen mass fraction on H, D and E are nonmonotonic. The nature of the obtained dependencies is very complex and not clear from the point of view of physics. The authors do not comment on this, while this is very important. Why should I trust these incomprehensible curves? Maybe the fact is that the solution is unsteady. Then it can contribute and influence the curves in the figures 12, 14, and 16?

Response: We sincerely appreciate the valuable comments, which has important guiding significance to our future studies. Firstly, the influence of the forcing duct outlet position, exhausting duct inlet position, and extraction pressure ratio on the oxygen mass fraction is not monotonous. The influence of these factors on the oxygen distribution is complex and first affect the flow characteristics of air flow in the roadway, thus affecting the oxygen distribution (modify in 445-447 line). Then, we used the same numerical model of roadway and ventilation system as the previous study, including the setting of some boundary conditions and the selection of turbulence model, and verified the accuracy of the numerical simulation results through experiments in the previous study [2], so this numerical simulation is reliable. Finally, this paper is mainly a theoretical and feasibility analysis of the optimization of the oxygen supply duct outlet form and the influence of the hybrid ventilation on the stability of oxygen enrichment effect, and found that the air duct location of the ventilation system has a great influence on the oxygen enrichment effect, so the use of oxygen supply device should be considered in combination with the ventilation system. We will further optimize the oxygen supply duct outlet form and further study the complex relationship of the influence of these factors on oxygen distribution from the perspective of fluid mechanics and other physics, employing laboratory experiments based on the numerical simulation results of this paper. Thank you again for your valuable advice.

Round 2

Reviewer 1 Report

The article with the observations indicated by the authors is suitable for publication.

Author Response

Thank you very much for your comment!

Reviewer 3 Report

The authors did a good job on the manuscript, but some of the comments are still not resolved. Below are my questions in response to my previous remarks:

(1) - the authors response is accepted.

(2) - it is necessary to write in the manuscript about the prismatic boundary layer and its parameters.

(3) - the authors response is accepted.

(4) - in the indicated place of the manuscript, the authors corrected the mistake, but in other places of the manuscript there is still confusion in the terms "grid", "mesh" and "cell". 

(5) - I noticed that the equations of continuity, Navier Stokes and concentration transfer appeared. But they are misspelled. There are no scalar multiplications, the velocity char is missing, the terms for diffusion are written incorrectly, the tensor is erroneously called a vector. No decoding of the abbreviation N-S.

Equation (3) is especially strange. Are you using a 2D model? And why divide by density? What then is the dimension of the diffusion coefficient?

It seems that the authors of the manuscript do not know the basics of computational fluid dynamics.

(6) - the authors response is accepted.

(7) - the answer is accepted. But in equations (1)-(3) the velocity vector U is denoted differently. Please, fix this.

(8) - the authors response is accepted.

(9) - the authors response is accepted.

(10) - it is not enough just to say that the model has been verified according to the experimental data. Moreover, I did not understand what kind of previous article [2] you are talking about. So, I don't know where to find proof of model verification.

In any case, a physical explanation is needed for the nonmonotonic dependences observed in numerical simulations.

Author Response

This manuscript is a resubmission of an earlier submission. The following is a list of the peer review reports and author responses from that submission.

Round 1

Reviewer 1 Report

some words are used improperly. "oxygen-enriched" should be "oxygen-enriching". "forcing duct" should be "supply duct" to match "exhaust duct".

The title "Optimization” is improper, since not many parameters were tested for an optimal result.

The title should be "Study on the Ventilation Scheme for Safe Working in the  Blind Heading Space of a Plateau Mine"

Reviewer 2 Report

The authors did not mention the experimental results in the abstract. Please briefly explain the experimental method and data in the abstract.

"Therefore, the effect of oxygen supply is not affected by altitude." The conclusion is not a general conclusion. The authors must rewrite the sentence in order to avoid misconceptions. For instance, "at the optimal ventilation scheme, the influence of altitude on the oxygen supply is not significant"

"the research method of oxygen distribution law is single" What do the authors mean? Please explained in detail to avoid a vague meaning.

"There are three types of basic auxiliary ventilation systems" Please explain using figures for a better understanding.

"the single use of exhaust ventilation will easily to cause the negative pressure state in the blind heading" Please add the reference to support this statement.

"while single forced ventilation will lead to a wide range of smoke diffusion and spread in the roadway along with the airflow" Please add the reference to support this statement.

"the hybrid ventilation mode" The authors must state that the hybrid ventilation mode is a combination of the exhaust and forced ventilation.

"The roadway model was established in FLUENT software" FLUENT is a CFD software, not a CAD software. Which FLUENT version did the authors use?  Now, FLUENT can be found in the ANSYS software. How to generate the mesh? Did the authors use ANSYS meshing? If yes, please state it in the manuscript. 

Please show us the mesh by figure. How many cells? What is the mesh type? hexahedral? tetrahedral? or ..?

Please do a mesh independent study.

In equation 1, it is written S. But it is written S0 in the text. Please revise.

"roadway (blind heading face), m2" Please revise, i.e., xx m2

Please describe the BCs by a table.

What a turbulence model did the authors use? Did authors try differents many turbulence models? Why did the authors choose that turbulence model, e.g., k-epsilon. 

Section 3 should be a 3. Numerical Simulation Results

"As can be seen from picture 4" picture change to figure.

Figure 4, case 3 has a reverse color plot.  Please revise.

Suddenly, the experimental results appear in section 3.3. The authors must describe the experimental methodology in the previous section. Otherwise, how to believe the experimental data is real data?